# Why people vote for thin-centred ideology parties? A multi-level multi-country test of individual and aggregate level predictors

**Hüseyin Çakal** [1]*, **Yasin Altınışık**[2], **Ömer Gökcekuş**[3], **Ertugrul Gazi Eraslan**[1]

**1** School of Psychology, Keele University, Keele, United Kingdom, **2** Sinop Universitesi, Sinop, Turkey, **3** Seton Hall University, South Orange, New Jersey, United States of America

* h.cakal@keele.ac.uk

**Citation:** Çakal H, Altınışık Y, Gökcekuş Ö, Eraslan EG (2022) Why people vote for thin-centred ideology parties? A multi-level multi-country test of individual and aggregate level predictors. PLoS ONE 17(3): e0264421. https://doi.org/10.1371/journal.pone.0264421

**Data Availability Statement:** We have now made the data publicly and freely available at: (https://osf.io/3epmh/).

**Funding:** The author(s) received no specific funding for this work.

## Abstract

The present research investigates the individual and aggregate level determinants of support for thin-centred ideology parties across 23 European countries. Employing a multilevel modelling approach, we analysed European Social Survey data round 7 2014 (N = 44000). Our findings show that stronger identification with one's country and confidence in one's ability to influence the politics positively but perceiving the system as satisfactory and responsive; trusting the institutions and people, and having positive attitudes toward minorities, i.e., immigrants and refugees, negatively predict support for populist and single issue parties. The level of human development and perceptions of corruption at the country level moderate these effects. Thus, we provide the first evidence that the populist surge is triggered by populist actors' capacity to simultaneously invoke vertical, "ordinary" people against "the elites", and horizontal, "us" against "threatening aliens", categories of people as well as the sovereignty of majority over minorities. These categories and underlying social psychological processes of confidence, trust, and threats are moderated by the general level of human development and corruption perceptions in a country. It is, therefore, likely that voting for populist parties will increase as the liberally democratic countries continue to prosper and offer better opportunities for human development. Stronger emphasis on safeguarding the integrity of the economic and democratic institutions, as our findings imply, and preserving their ethical and honest, i.e., un-corrupt, nature can keep this surge under check.

## Introduction

In this study, we focus on the rise of *populism* in *liberal democracies* in Europe. We define *populism* as a two-dimensional framework of ideas [1], that includes but not limited to simultaneous invocation of vertical, "ordinary" people against "the elites", and horizontal, "us" against "threatening aliens", categories of people as well as the sovereignty of majority over minorities, and overprotective anti-institutionalism that purports to protect economic and cultural rights of "people" against threats from top, bottom, or from outside [1–3]. Our operationalisation of populism as a "two-dimensional" framework of inequality (vertical) and difference

**Competing interests:** NO authors have competing interests.

(horizontal) allows us to go beyond the radical right and left divide, and to include not only right or left wing populist parties but also "single issue" parties [4] that focus on an "all compassing issue" without an ideological basis across the political continuum [5]. *Liberal democracies* on the other hand are political systems that protect the civil, political, and economic rights of their citizens and provide access to educational, economical, and social opportunities on the basis of equality within an ethical and fair system [6]. In the current research, we explore how a host of individual and country level processes lead to more support for thin ideology parties. At the individual level we focus on subjective experiences resulting from human tendency to categorize their social environment into groups [7–9], ensuing threats and status concerns. We consider these processes as embedded in perceived vertical inequalities (dis-trust in institutions, satisfaction with the system). In addition, we also investigate (national identification), attitudes toward immigrants, and attitudes toward support for policies benefitting refugees embedded in perceived horizontal differences. We then focus on how these individual processes interact with country level factors such as human development opportunities (HDI) and corruption perceptions influence populist voting. such as attachment to one's country (national identification).

Thus, we contribute to the ongoing debate on populism by showing that support for populist parties is driven by a host of seemingly innocuous social psychological mechanisms triggered by perceived inequalities, i.e., trust, system perceptions, and perceived differences, e.g., national identification attitudes toward *new minorities* [10]. We then show how aggregate level proxies of these differences, corruption perceptions and the level human development moderate both vertical and horizontal dimensions. By doing so, we connect political scientific research on populism with social psychological processes and provide a general account of the factors that motivate populist support across the political spectrum.

## Why do people support thin-centred ideology parties?

One particular line of research shows that economic anxieties, perceptions of multiculturalism, and increasing numbers of immigrants and refugees motivate support for the populist parties [1, 2, 11]. The so called *economic anxiety thesis* [12] posits that certain segments, the low paid manual working classes, of the societies in the developed world perceive themselves as left behind in the new liberal economies of a globalized world. These segments also hold economic and political elites as responsible for their plight and turn to populist parties and leaders who exploit their resentment against the system.

This prediction is based on the assumption that the post-industrial societies have gone through drastic changes that transformed the labour politics and society in general. Increasing reliance on knowledge as a commodity, automation in the workplace, and the collapse of the manufacturing based economies coupled with the influx of foreign capital, goods, and cheaper labour in the form of migrants and refugees have made "the ordinary people" or "the silent majority", low-paid manual labourers, more vulnerable to economic ups and downs. This sense of vulnerability in turn breeds resentment against the state and motivates support for populist actors who appear to defend *the silent masses* against the so-called elite and promise to bring back the *glory of the old days* [12]. In a similar vein, *cultural backlash hypothesis* [13, 14] predicts that post-materialist generations growing up in the economically and physically secure welfare democracies have had access to better and more educational opportunities and supported a range of improvements in human rights, environmental protection, and gender equality. These changes threaten the core traditional values of older and less educated generations and trigger an authoritarian reaction [14] making them more amenable to the exploits of

the populist parties and leaders [2, 15, 16]. In fact, there is ample evidence on how individuals respond to such threats.

For instance, previous research on intergroup relations shows that individuals' perceptions of their group as relatively deprived and disadvantaged compared to their group's past or future conditions motivates discrimination [17, 18] against whom they attribute the responsibility of their present or anticipated disadvantage. Similarly, when individuals perceive their resources, lifestyles, and their belief systems as threatened they react by discriminating against the sources of threat [19, 20].

Taken together, the economic anxiety and cultural backlash hypotheses, i.e. *the losers of globalization*, predict that populist parties with thin-centred ideologies would garner the most support from older, less educated, less paid individuals living in the post industrialist countries with higher net migration rate with stuttering economies. Recent evidence however has been inconclusive and supported only one dimension of *the losers of globalization* thesis. Economically less well-off individuals tend to vote for radical left and right parties when their country's' economic performance is favourable and the level of immigration is modest [15].

An alternative line of research argues that it is not a general sense of relative deprivation but a sense of relative wealth that enhances populist support [21, 22]. According to *the wealth paradox* hypothesis as economy performs well and the society prospers the economically well-off display the most xenophobic and anti-immigrant attitudes. This is partly motivated by fear of future deprivation and loss of wealth, and partly by a sense of relative gratification resulting from in-group's attractiveness [23–25]. This sense of gratification in turn intensifies the allure of the in-group, and triggers fear of possible loss of some privileges as well as an experience of guilt about them. Together, both fear and guilt motivate proactive discrimination against the outgroups, immigrants and refugees, as the economically well-off anticipate the immigrants and refugees to challenge the disparities [22]. The anticipation of challenge then drives the support for populist parties who thrive on these concerns. Although scarce, empirical support for "the wealth paradox" shows that those who experience higher levels of gratification display the strongest opposition to immigration [22].

In the present research, we argue that these seemingly contradictory arguments result from populist capacity to construct and reinforce identities on the basis of perceived vertical inequalities, "us" versus "elites", and perceived horizontal differences, "us" versus "aliens". Social psychological research on mobilization and political activism shows that individuals are motivated to engage in political action when they perceive themselves as deprived, relatively, compared to the other outgroups when they perceive their privileges and their way of life as threatened [26], or when they subjectively think that they have the capacity to change their actual or future disadvantage [27]. More specifically, because populist narrative delineates group boundaries and emphasizes both perceived disadvantages between "the ordinary people" and the "corrupt politicians and elites" (vertically) on one hand, and perceived threats from "aliens" (horizontally) on the other, it goes beyond the right versus left categories. Right and right leaning populist actors use this framework to project the elites (vertical) and the foreigners, e.g. refugees, immigrants (horizontal) as enemies of the ingroup, the ordinary people [7, 28, 29]. Left and left leaning populist actors on the other hand. Thus, by invoking both vertical inequalities and horizontal differences across a wide range of categories, populist narrative categorizes people into "us" versus "them" and projects the "others" as threats [7]. Although crucial to mobilization, this collective identity is not enough to get people to support a particular party or a leader. Individuals will only vote for those actors who appeal to their discomfort. The populist actors to *employ* this identity that they crafted to mobilize "us" against those who are responsible for the hardships and threats that "we" are experiencing [7, 30]. Two factors could facilitate these processes at the country level. First, and in line with the

*wealth paradox*, one could argue that more and better access to educational, economic, and social opportunities in a country results in improved welfare of the citizens in that country and gives them an inflated sense of empowerment. At the same time, when this sense of empowerment is accompanied by sense of "us "delineated by vertical deprivation, current or future, sense of dissatisfaction with the system, and distrust in the institutions and horizontal threats from "aliens" turn to the populist parties.

Second, "The corruption" rhetoric that the populist actors employ portrays the establishment as corrupt and blames it for the hardships the ordinary people are going through. Once the perceptions of corruption are in place, the populist actors use these perceptions to push forward several conclusions. First and foremost, they emphasize that the establishment is not with the ordinary people which now cunningly include both the people and them as defenders of the ordinary people's rights. Second, they also emphasize that the establishment work against the will of ordinary people and therefore it is not to be trusted.

This emphasis on ingroup and outgroup as "*the establishment*", brings forward the issue of trust and system responsiveness. In general, trust is the belief that the other party will not exploit the vulnerabilities that the actors have [31, 32]. We define trust as the perception that state institutions will stay true to deliver what is expected from them [33, 34]. As such, trust based on iterative interactions with the state institutions and other parties. Institutionalized trust is also grounded in what we call the "non-rivalry" assumption that the state institutions and individuals are not competitors to each other [35]. This assumption resonates with the fundamental structure of the liberal economies that provide and protect civil, political, and economic rights of their citizens, and implies that higher levels of trust in the public institutions is an indicator of perceptions of public institutions as well-performing units [36]. Creating the in-group versus the outgroup distinction challenges the non-rivalry assumption and foments distrust between the in-group, the ordinary people, and the out-group, the establishment. In our analysis, we include both institutional trust and generalized trust.

The outline we provide above suggest that support for populist parties is a co-product of core social psychological processes, i.e., social identity, trust, satisfaction with the system, perceptions of efficacy, attitudes toward immigrants and refugees, and structural factors, level of affluence and the corresponding opportunities for human development as well corruption perceptions. Thus, we incorporate these country level dynamics to our conceptual model and deduce several testable hypotheses that might explain the rise of populist parties with a thin ideological base.

At the individual level, we draw from social psychological literature on social identity [37] and collective action [27]. We hypothesize that stronger identification with one's country (CI) and confidence in one's ability to take active part in politics (PC) grounded in this identification would be positively associated with support for populist parties (Hypothesis 1a and Hypothesis 1b, H1a and H1b respectively). Conversely, perceived system responsiveness (PSR), dimensions of trust, generalized (TR) and institutional (ITR); general sense of satisfaction (SAT), attitudes toward immigrants (AtI) and support for policies benefiting refugees (AtR) would be negatively associated with support for populist parties (Hypothesis 2a, Hypothesis 2b, Hypothesis 2c, Hypothesis 2d; Hypothesis 2e; and Hypothesis 2f; H2a; H2b, H2c, H2d, H2e, and H2f, respectively)

At the country level, our core prediction is that the effect of individual level social psychological variables on populist voting will be influenced by two aggregate level processes, the level of human development in a country, operationalized as Human Development Index [HDI, 38] and perceptions of corruption measured by Corruption Perception Index [CPI, 39]. Thus, we expect HDI to positively moderate the effects of country identification, political confidence, perceived system responsiveness, and trust in institutions and individuals, and

attitudes toward immigrants and refugees on populist support. At higher levels of HDI both the positive effects of country identification and political confidence one hand (Hypothesis 3a; H3a) and the negative effect of perceived system responsiveness, dimensions of trust and attitudes toward immigrants and refugees on populist voting (Hypothesis 3b; H3b) on populist voting on other would be stronger. Conversely, we expect the negative effect general satisfaction with the system on populist voting to be reversed as HDI increases (Hypothesis 3c, H3a).

Our expectations for the moderating role of corruption perceptions at the country level differ slightly then that of human development opportunities. We expect the positive effect of country identification on populist voting and political confidence to be reversed (Hypothesis 4a and 4b; H4a, 4b, respectively). As the individuals perceive the establishment as more corrupt the country identification will have a negative effect on populist voting while political confidence will have a positive effect on populist voting (H4b). On the other hand, we expect the negative effect of perceived system responsiveness, dimensions of trust, and general satisfaction to disappear when they perceive the establishment as corrupt (Hypothesis 4c; H4c) if not reversed. More specifically, when individuals perceive the establishment as corrupt this might nullify the effect of these variables on populist voting as they are grounded in perceptions of institutions' as being able to deliver. Last but not least, we expect the negative impact favourable attitudes toward immigrant and refugees to be enhanced to the extent that they perceive the establishment as corrupt (Hypothesis 4d, H4d).

To test these hypotheses, we followed a model building strategy and employed multilevel logistic regression which is a convenient analytical strategy when individual data is nested in some higher-level units, in our case in countries. Our decision is to depart from unidimensional indicators of wealth and inequality (namely GDP *per capita*, Gini coefficient, net migration rate, social welfare expenditure by country, respectively) employed by previous research [15] and focus instead on the subjective experience at individual and aggregate level and a multidimensional measures of human development (HDI) and corruption perceptions.

## Method

### Data and measures

We tested our hypotheses using ESS data (Round 7) across 23 European countries. Our individual data is nested in countries and our dependent variable is binary, i.e., voted for populist or single-issue parties in the last election (see online Appendix in S1 File for the list of the parties we included in the analyses). In addition to our social psychological variables at the individual-level and aggregate variables at the country-level, i.e., HDI and CPI, we also included demographic variables for control purposes.

**Dependent variable.** Our dependent variable is whether individuals voted for populist or single parties or not. ESS data contains information on voting during the last elections. We created a binary variable and coded individual responses 0 (voted for a non-populist party) and 1 (voted for a populist party, see the list of the parties we included in our analysis in the S1 File). We employ the most recent classification scheme used by Pew Research Center in their report European Public Opinion Three Decades After the Fall of Communism [40].

**Demographic variables.** All of demographic variables are categorical, that is, gender (G: 0 = male and 1 = female), paid work (P 0 = not in paid work and 1 = in paid work), and the highest level of education (E1 = less than lower secondary education, 2 = lower secondary education completed, 3 = upper secondary education completed, 4 = post secondary non-tertiary education completed, and 5 = tertiary education completed).

**Individual level variables.** *Country identification* (CI) is measured on a 11-point (0, *not at all emotionally attached*; 10, *very emotionally attached*). *Political confidence is* measured by two

items (r = .79, *p* < .001; PC 'able to take active role in politics' and 'confident in own ability to participate in politics'; 0, *not at all*; 5, *a great deal*). We assessed *perceived (political) system responsiveness* (PSR) by two items (*r* = .60, *p* < .001): 'How much would you say the political system in [country] allows people like you to have a say in what the government does/to have an influence on politics?' (1, *not at all*; 5, *a great deal*). We used three items (α = .76) to measure *generalized trust* (TR 0, *you can't be too careful/most people try to take advantage of me/ people mostly look out for themselves*; 10, *most people can be trusted/most people try to be fair/ people mostly try to be helpful*). Similarly, we used five items (α = .88) to measure *institutional trust* (ITR), the level of trust individuals have in the institutions of their country, i.e., the parliament, the legal system, the police, the politicians, and the political parties (0, *no trust at all*; 10, *complete trust*).

We used six items (α = .79) to measure the *general level of satisfaction* with different aspects of the system on an 11-point scale (SAT 0, *extremely dissatisfied/bad*; 10, *extremely satisfied/ good*). The participants indicated their level of satisfaction with life as a whole, with the economy in country, national government, the way democracy works, education, and health services in their country. We used three items to measure (α = .87) attitudes toward immigrants (AtI) on an 11-point scale (0, *bad for the economy/culture/ social life*; 10, *good for the economy/ culture/ social life*). In a similar vein, three items were employed as a proxy to attitudes toward refugees (AtR α = .61): "The government should be generous in judging people's applications for refugee status", 'Most refugee applicants not in real fear of persecution own countries' and 'Granted refugees should be entitled to bring close family members' (1, *agree strongly*; 5, *disagree strongly*). We reverse coded the first and the third item so cumulatively higher values represent more support toward policies benefitting refugees. Although we have no specific hypothesis regarding *political orientation*, the ESS survey has an item (11-point scale ordinal variable 0, *left*; 10 *right*) so we included this item in our analyses as a control variable.

**Country level variables.** We measure *the level of human development* with human development index (HDI) prepared by the United Nations Development Agency [41]. HDI is based on Amartya Sen's [42] conceptualization of development as a tool to improve human experience by expanding individuals capacity to be healthier, more knowledgeable, and to be able to be a fully active, civically and politically, member of the society. It is a composite index that represents the three dimensions of development: a) longevity, b) knowledge, and c) access to resources [43, 44]. These three dimensions then are incorporated to rank countries on a scale that ranges from 0 (lowest level of human capacity for development) to 1 (highest level of human capacity for human development).

We measured *the perceived level of corruption* with corruption perception index (CPI). As a composite index, CPI incorporates perceptions of various forms of corruption, e.g. bribery, diversion of public funds, or nepotism in the civil service, as well existence of mechanisms to combat corruption [39]. The index ranges from 0 (very high level of corruption) to 100 (no corruption). Taken together, both variables allow us to gauge the capacity of development that each country offers to its citizens as well as the perceptions of the corruption as determinants of support for populist parties as predicted by the ideational perspective. We report means, standard deviations, and correlations between our variables of interest in Table 1.

## Analysis and results

We fit a series of models to predict the log odds of populist voting to evaluate our hypotheses. We also include odds ratios and their confidence intervals to interpret the results for each hypothesis under consideration and to inspect whether the assumptions of the multilevel

**Table 1. Descriptive statistics and correlations of the variables included in the models.**

|  | HDI | CPI | PO | CI | PC | TR | ITR | SAT | AtI | AtR | PSR |
|---|---|---|---|---|---|---|---|---|---|---|---|
| **HDI** |  |  |  |  |  |  |  |  |  |  |  |
| **CPI** | 0.82 |  |  |  |  |  |  |  |  |  |  |
| **PO** | -0.03 | 0.01 |  |  |  |  |  |  |  |  |  |
| **CI** | -0.02 | 0.01 | 0.15 |  |  |  |  |  |  |  |  |
| **PC** | 0.22 | 0.18 | 0.04 | 0.02 |  |  |  |  |  |  |  |
| **TR** | 0.29 | 0.31 | 0.01 | 0.11 | 0.16 |  |  |  |  |  |  |
| **ITR** | 0.28 | 0.32 | 0.06 | 0.17 | 0.19 | 0.46 |  |  |  |  |  |
| **SAT** | 0.30 | 0.32 | 0.15 | 0.19 | 0.12 | 0.42 | 0.68 |  |  |  |  |
| **AtI** | 0.21 | 0.28 | -0.15 | 0.01 | 0.24 | 0.33 | 0.33 | 0.30 |  |  |  |
| **AtR** | 0.12 | 0.16 | -0.24 | -0.04 | 0.16 | 0.21 | 0.19 | 0.11 | 0.56 |  |  |
| **PSR** | 0.27 | 0.29 | -0.01 | 0.05 | 0.42 | 0.30 | 0.49 | 0.43 | 0.32 | 0.22 |  |
| **Mean** | 0.90 | 70.83 | 5.16 | 8.02 | 2.31 | 5.76 | 5.10 | 5.78 | 5.44 | 3.07 | 2.36 |
| **(SD)** | (0.03) | (14.11) | (2.29) | (2.02) | (0.97) | (1.77) | (1.93) | (1.57) | (2.16) | (0.84) | (0.84) |

logistic regression models that are used to evaluate our hypotheses are reasonably satisfied by the ESS data.

An important condition for adapting a multilevel approach and employing aggregate level variables is to empirically demonstrate that there is between level variance across our variables. Thus, the intercept-only model can be used to determine whether the use of multilevel logistic regression modelling on predicting the outcome populist voting is superior to that of standard (single-level) logistic regression modelling. For this purpose, we calculated an intra-class correlation coefficient (ICC) ranging from 0 to 1 based on the proportion between the between-country variation and the total variation (i.e., the sum of the between-country variation and the within-country variation for populist voting). By utilizing a simulation based approach [45] which involves the estimated values of the overall intercept (i.e., $\hat{\beta} = -1.53$) and the random intercept variance (i.e., $\hat{\tau}^2 = 1.88$) for the intercept-only model, we obtained the ICC as 0.23. This means that 23% of the effects on the probability of populist voting for a participant in the survey come from between-country dissimilarities, while 77% of these effects come from within-country dissimilarities. Thus, we conclude that the use of an intercept-varying multi-level logistic regression model in predicting the outcome populist voting is more reasonable than that of a standard (single-level) logistic regression (see Tables 2 and 3). Additional analysis showed that the model that contains both individual-level and country-level main effects, $M_2$, fit the data better than the intercept-only model, $M_0$, and the model containing only individual-level main effects, $M_1$, as indicated by lower AIC and BIC values, and deviance values.

**Predicting the log odds of populist voting.** We utilize the method of maximum likelihood to estimate model parameters and their standard errors when evaluating our hypotheses. We report the findings with respect to the effects of demographic variables, individual level social psychological variables, aggregate level variables, and the interactions of social psychological variables and aggregate variables on populist voting, incrementally, in Table 3 via $M_{31}$, $M_{32}$, and $M_{33}$. For economics of space and clarity however we focus on $M_{33}$, as it enables us to test all four hypotheses simultaneously (See Table 3 and S1 File for additional information on the analyses).

From a demographic point of view, males (β = -.17, SE = .04, $p < .001$), those who completed upper secondary (β = .22, SE = .09, $p < .05$, and who are right leaning (β = .15, SE = .02, $p < .001$) tend to vote more for the populist parties.

**Table 2. Intercept-only model and models containing only main effects.**

| Main effects (fixed) | $M_0$ | | $M_1$ | | $M_2$ | |
|---|---|---|---|---|---|---|
| **Individual-level** | β | SE | β | SE | β | SE |
| (Intercept) | -1.53*** | .29 | -1.60*** | .31 | -1.56*** | .29 |
| gender | | | -.17*** | .04 | -.17*** | .04 |
| paidwork | | | -.07 | .04 | -.07 | .04 |
| education2 | | | .04 | .10 | .04 | .10 |
| education3 | | | .22* | .09 | .22* | .09 |
| education4 | | | .19 | .11 | .19. | .11 |
| education5 | | | .03 | .09 | .03 | .09 |
| PO | | | .15*** | .02 | .15*** | .02 |
| PC | | | .05* | .02 | .05* | .02 |
| CI | | | .03. | .02 | .03. | .02 |
| PSR | | | -.09*** | .02 | -.09*** | .02 |
| TR | | | -.05* | .02 | -.05* | .02 |
| ITR | | | -.15*** | .03 | -.15*** | .03 |
| SAT | | | -.23*** | .03 | -.23*** | .03 |
| AtI | | | -.07** | .02 | -.07** | .02 |
| AtR | | | -.16*** | .02 | -.16*** | .02 |
| **Country-level** | | | | | | |
| HDI | | | | | 1.06* | .46 |
| CPI | | | | | -0.90* | .46 |
| **Random effects** | $\sigma^2_{\tau_{0j}}$ | $\sigma_{\tau_{0j}}$ | $\sigma^2_{\tau_{0j}}$ | $\sigma_{\tau_{0j}}$ | $\sigma^2_{\tau_{0j}}$ | $\sigma_{\tau_{0j}}$ |
| (Intercept) | 1.88 | 1.37 | 1.96 | 1.40 | 1.60 | 1.27 |
| **Model fit indices** | | | | | | |
| AIC | 18192.6 | | 17489.7 | | 17488.8 | |
| BIC | 18208.5 | | 17624.4 | | 17639.3 | |
| Deviance | 18188.6 | | 17455.7 | | 17450.8 | |

When all variables of interest and their interactions were entered into the model ($M_{33}$) political confidence (PC; β = .07, SE = .02, $p$ < .001) but not country identification had a positive effect on populist voting. As expected, however, dimensions of trust, generalized (TR; β = -.05, SE = .02, p < .05) and institutional (ITR; β = -.17, SE = .03, $p$ < .001), perceived system responsiveness (PSR; β = -.09, SE = .03, $p$ < .001), and overall satisfaction with the system (SAT; β = -.24, SE = .03, $p$ < .001). Last but not least, positive attitudes toward immigrants (AtI β = -.13, SE = .03, $p$ < .001) and toward refugees (AtI β = -.13, SE = .02, $p$ < .001) were negatively associated with populist voting.

At the aggregate level, results show that there is a strong positive effect of HDI (β = 1.23, SE = .48, $p$ < .001) on populist voting. As the level of human development in a country increases so does the populist voting. The effect of CPI was also significant and in the expected direction. Perceiving the system as ethical and honest, i.e. higher values on the CPI is associated with decreased support for populism (CPI; β = -1.10, SE = .48, $p$ < .05).

Looking at the moderating effects of HDI, the results show that when HDI increased, political confidence (PC:HDI) did not longer predict populist voting but country identification did (CI:HDI, β = .11, SE = .04, $p$ < .01; Fig 1A). In countries that rank higher on HDI the more people identify with their countries the more they support populist parties. The negative effect of generalized trust did not change as HDI increased (TR:HDI, β = -.18, SE = .04, $p$ < .001;

**Table 3. Models containing cross-level (two-way) interactions.**

| (Fixed effects) | Model $M_{31}$ | | | Model $M_{32}$ | | | Model $M_{33}$ | |
|---|---|---|---|---|---|---|---|---|
| **Individual-level main effects** | **β** | **SE** | | **β** | **SE** | | **β** | **SE** |
| (Intercept) | -1.55*** | .29 | | -1.58*** | .30 | | -1.57*** | .30 |
| Gender | -.18*** | .04 | | -.17*** | .04 | | -.17*** | .04 |
| Paidwork | -.08 | .04 | | -.07 | .04 | | -.07 | .04 |
| education2 | .05 | .10 | | .05 | .10 | | .06 | .10 |
| education3 | .23* | .09 | | .21* | .09 | | .22* | .09 |
| education4 | .20 | .11 | | .19 | .11 | | .20. | .11 |
| education5 | .05 | .09 | | .04 | .09 | | .05 | .09 |
| PO | .16*** | .02 | | .15*** | .02 | | .15*** | .02 |
| CI | .02 | .02 | | .03 | .02 | | .01 | .02 |
| PC | .06** | .02 | | .05* | .02 | | .06** | .02 |
| PSR | -.10*** | .03 | | -.08*** | .03 | | -.09*** | .03 |
| TR | -.05* | .02 | | -.06** | .02 | | -.05* | .02 |
| ITR | -.21*** | .03 | | -.12*** | .03 | | -.17*** | .03 |
| SAT | -.22***· | .03 | | -.27*** | .03 | | -.24*** | .03 |
| AtI | -.08** | .02 | | -.14*** | .03 | | -.13*** | .03 |
| AtR | -.15*** | .02 | | -.13*** | .03 | | -.13*** | .03 |
| **Country-level main effects** | | | | | | | | |
| HDI | 1.17* | .47 | | 1.27** | .48 | | 1.23* | .48 |
| CPI | -1.02* | .47 | | -1.13* | .48 | | -1.10* | .48 |
| **Cross-level interactions** | | | | | | | | |
| CI:HDI | .13*** | .04 | SAT:HDI | .41*** | .04 | CI:HDI | .11** | .04 |
| PC:HDI | -.09* | .04 | AtI:HDI | -.01 | .04 | PC:HDI | -.06 | .04 |
| PSR:HDI | .11** | .04 | AtR:HDI | -.15*** | .04 | PSR:HDI | .08 | .04 |
| TR:HDI | -.15*** | .04 | SAT:CPI | -.35*** | .04 | TR:HDI | -.18*** | .04 |
| ITR:HDI | .32*** | .04 | AtI:CPI | -.15*** | .04 | ITR:HDI | .13** | .05 |
| CI:CPI | -.11*** | .03 | AtR:CPI | .08 | .04 | SAT:HDI | .34*** | .05 |
| PC:CPI | .11** | .04 | | | | AtI:HDI | .01 | .04 |
| PSR:CPI | -.13** | .04 | | | | AtR:HDI | -.15*** | .04 |
| TR:CPI | .08* | .03 | | | | CI:CPI | -.11** | .03 |
| ITR:CPI | -.34*** | .04 | | | | PC:CPI | .11** | .04 |
| | | | | | | PSR:CPI | -.09* | .04 |
| | | | | | | TR:CPI | .13*** | .04 |
| | | | | | | ITR:CPI | -.18*** | .05 |
| | | | | | | SAT:CPI | -.23*** | .05 |
| | | | | | | AtI:CPI | -.16*** | .04 |
| | | | | | | AtR:CPI | .08 | .04 |
| **(Random effects)** | $\sigma^2_{\tau_{0j}}$ | $\sigma_{\tau_{0j}}$ | | $\sigma^2_{\tau_{0j}}$ | $\sigma_{\tau_{0j}}$ | | $\sigma^2_{\tau_{0j}}$ | $\sigma_{\tau_{0j}}$ |
| (Intercept) | 1.69 | 1.30 | | 1.78 | 1.33 | | 1.77 | 1.33 |
| **Model fit indices** | | | | | | | | |
| AIC | 17336.3 | | | 17258.0 | | | 17216.7 | |
| BIC | 17566.2 | | | 17456.1 | | | 17494.0 | |
| Deviance | 17278.3 | | | 17208.0 | | | 17146.7 | |

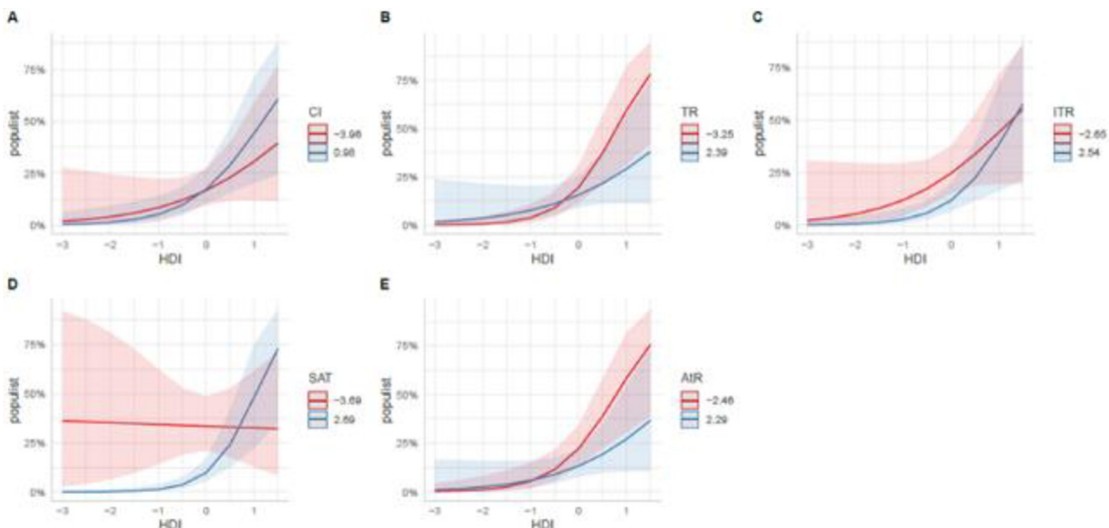

**Fig 1. A-E.** Plots showing the moderating effect of HDI on CI, TR, ITR, SAT, and AtR, respectively. Pink and blue lines show the probabilities of populist voting with increasing values of HDI for low and high values of attitudinal variables, respectively. Pink and blue areas display the corresponding confidence intervals.

Fig 1B). Surprisingly however HDI reversed the negative effect of institutional trust (ITR:HDI, β = .13, SE = .05, $p < .05$; Fig 1C) on populist voting. The negative effect of institutional trust at the individual level changes direction. In countries that rank higher on HDI, the more people trust in the institutions of their country the more they support populist parties.

In a similar vein, providing the strongest support for the wealth paradox, HDI reversed the negative effect of system satisfaction on populist voting. As HDI increased effect of system satisfaction become positive (β = .34, SE = .05, $p < .001$; Fig 1D). In countries that rank higher on HDI, overall satisfaction with the system is associated with more support for populist parties. The negative impact of attitudes toward refugees (AtR) on populist voting at the individual did not change (AtR:HDI; β = -.15, SE = .04, $p < .001$; Fig 1E) as HDI increased but the negative impact of attitudes toward immigrants on populist voting was reversed but did not reach the level of significance.

With regards to the moderating effect of CPI on social psychological variables-populist voting paths (note that higher values on CPI shows that the system is perceived less corrupt) our findings show that CPI moderated the effect of CI on populist voting (CI:CPI, β = -.11, SE = .03, $p < .001$, Fig 2A). In countries where the system is perceived ethical and honest the more people identify with their country the less they support populist parties. In similar vein, we observed similar effects of CPI on the effects of other social psychological variables on populist voting. Surprisingly, the positive effect of PC on populist voting at the individual level did not change (PC:CPI, β = -.11, SE = .04, $p < .05$; Fig 2B) as CPI increased. Individuals' confidence in their ability to take active part in politics positively predict populist voting only when the system is perceived less corrupt. In similar vein, the negative effects of perceived system responsiveness (PSR) remained as significantly negative (Fig 2C).

Surprisingly, CPI also reversed the negative effects of trust dimensions on populist voting. Previously negative, the effect of generalized trust became positive (TR:CPI, β = .13, SE = .04, $p < .001$; Fig 2D), Interestingly, previously negative effect of institutional trust did not change (β = -.18, SE = .04, $p < .001$; Fig 2E). In countries where the system is perceived as ethical and honest, the more people trust others the more they supported populist parties. However, the

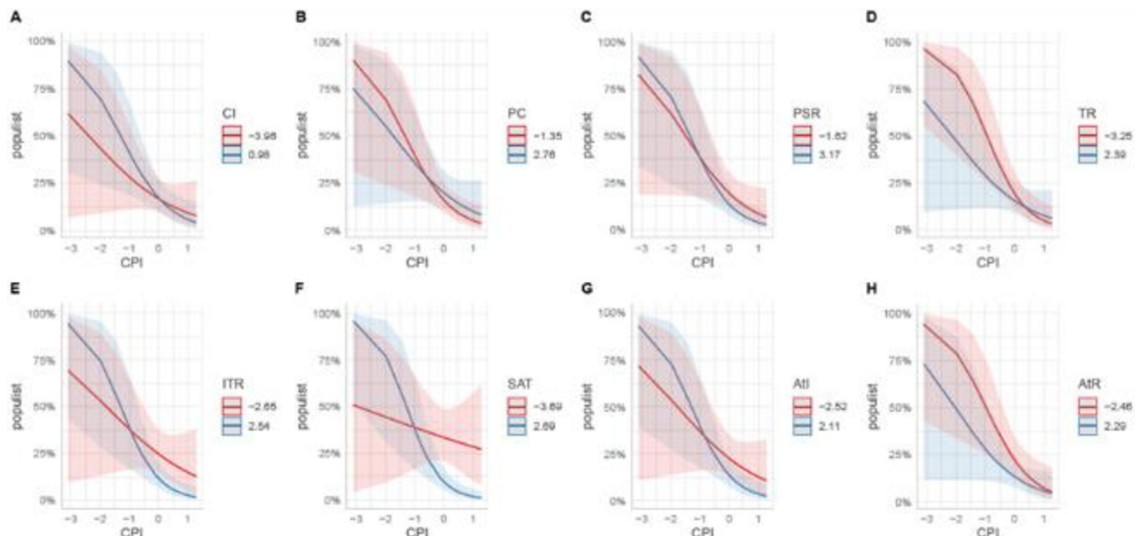

**Fig 2. Plots showing the moderating effect of CPI on CI, PC, PSR, TR, ITR, SAT, AtI, and AtR, respectively.** Pink and blue lines show the probabilities of populist voting with increasing values of CPI for low and high values of social psychological variables, respectively. Pink and blue areas display the corresponding confidence intervals.

more they trust institutions the less they support populist parties. We outline the implications of this difference in the general discussion.

Last but not least, as CPI increased, the negative effects of overall satisfaction and attitudes toward immigrants on populist voting did not change (Fig 2F and 2G) while the effect attitudes toward refugees (AtR:CPI) changed to negative but did not reach to statistical significance.

To increase confidence in our results and for ease of interpretation we conducted additional analyses of the odd ratios with Wald confidence intervals (Table 4). The odds ratios can be directly obtained by exponentiating the estimated values of the model parameters when predicting the log odds of the outcome populist voting. To illustrate, consider the effect of the predictor gender on the log odds of the outcome populist voting when evaluating hypothesis $H_1$ using model $M_{31}$, where $\hat{\beta} = -0.18$. Then, the odds ratio representing the association between the outcome populist voting and the predictor gender can be calculated as OR = $\exp(\hat{\beta})$ = exp (-.18) ≈ .84. The interpretation of this value and the other odds ratios should be made with reference to the value of 1. For example, since the confidence interval between the lower and upper bounds of this odds ratio [.78, .91] does not contain the value of 1, we conclude that there is a significant negative association between gender and populist voting. This means that the probability of populist voting (instead of mainstream voting) is approximately 1/.84 ≈ 1.19 times smaller for the females when compared to the reference category, i.e., males. Similarly, the odds ratios for the association between HDI human and populist voting is OR = $\exp(\hat{\beta})$ = exp(1.17) ≈ 3.22 and the confidence interval between the lower and upper bounds for this odds ratio [1.27, 8.09] does not contain 1. This means that the probability of populist voting (instead of mainstream voting) increases approximately 3.22 times by one unit increase in human development index. In contrast, the odds ratio between the outcome populist voting and the predictor CI does not indicate a significant association between these variables. That is, OR = $\exp(\hat{\beta})$ = exp(.02) ≈ 1.02 is almost equal to 1 and the confidence interval between the lower and upper bounds for this odds ratio (.98, 1.076 includes 1.).

**Table 4. The odds ratios and Wald confidence intervals between brackets for the models containing cross-level (two-way) interactions.**

| Individual-level main effects | Model $M_{31}$ | Model $M_{32}$ | Model $M_{33}$ |
|---|---|---|---|
| | Odds ratios (Wald CI) | Odds ratios (Wald CI) | Odds ratios (Wald CI) |
| (Intercept) | .21 (.12, .38) | .21 (.11, .37) | .21 (.11, .37) |
| gender | .84 (.78, .91) | .85 (.78, .91) | .84 (.78, .91) |
| paidwork | .93 (.86, 1.00) | .93 (.86, 1.01) | .93 (.86. 1.01) |
| education2 | 1.05 (0.87, 1.28) | 1.05 (.87, 1.28) | 1.07 (.87. 1.28) |
| education3 | 1.25 (1.05, 1.50) | 1.23 (1.03, 1.47) | 1.24 (1.03. 1.47) |
| education4 | 1.22 (0.99, 1.52) | 1.21 (.97, 1.50) | 1.22 (.97. 1.50) |
| education5 | 1.05 (.87, 1.26) | 1.04 (.87, 1.25) | 1.05 (.87, 1.25) |
| PC | 1.07 (1.02, 1.11) | 1.05 (1.01, 1.10) | 1.07 (1.01, 1.10) |
| PO | 1.17 (1.12, 1.22) | 1.16 (1.12, 1.21) | 1.16 (1.12, 1.21) |
| CI | 1.02 (.98, 1.06) | 1.03 (.99, 1.07) | 1.01 (.99, 1.07) |
| PSR | .91 (.86, .96) | .92 (.88, .97) | .91 (.88, .97) |
| TR | .95 (.90, .99) | .94 (.90, .98) | .95 (.90, .98) |
| ITR | .81 (.76, .86) | .88 (.84, .93) | .84 (.83, .93) |
| SAT | .80 (.76, .85) | .76 (.72, .81) | .79 (.72, .81) |
| AtI | .93 (.88,.97) | .87 (.83,.92) | .87 (.83, .92) |
| AtR | .86 (.82, .90) | .87 (.83, 0.92) | .88 (.83, 0.92) |
| **Country-level main effects** | | | |
| HDI | 3.22 (1.27, 8.09) | 3.55 (1.38, 9.15) | 3.41 (1.38, 9.15) |
| CPI | .36 (.14,.91) | .32 (.13, .84) | .33 (.13, .84) |

## Additional tests

To further increase our confidence in the findings, we conducted additional tests to investigate whether the results are biased due tomulticollinearity, strong correlation (s) between two or more predictors in a regression model. Although multicollinearity does not distort the reliability or power of the model as a whole [46], it may cause spuriously high standard errors of individual parameter estimates, and thus, undependable test statistics [47]. For example, it is a quite common scenario in the results of regression analysis where the F-test for a model indicates a significant overall effect of predictors on an outcome, while none of the t-tests in the same study demonstrate a considerable individual impact. Severe multicollinearity in the data may prevent researchers from diffrentiating individual impacts of predictors on the model outcome. This, in turn, mightcause interpretation problems.

Multicollinearity requires even more careful attention in logistic hierarchical modeling than the standard logistic regression, since it can occur not only for subject level variables, but also for aggregate level variables. As can be seen on the bottom panel of Table 1, only the correlation between aggregate level variables HDI and CPI (i.e., |0.82| > 0.7) is quite large indicating a possible multicollinearity problem in the data in line with the suggestion made in [48]. However, the high correlations among the predictors do not necessarily mean the data at hand suffer from the multicollinearity problem. Another diagnostic test for detecting multicollinearity is the variance inflation factor (VIF). This factor measures the inflation in the variance of the estimate for each regression coefficient in the model by taking into account the correlations among the predictors in the data. The VIFs that are larger than 3, 5, or 10 or the small values of its inverse, the tolerance (TOL) are considered to be a sign of multicollinearity problem in the data [49]. In addition to these values, we elaborate on two additional diagnostic measures

**Table 5. Multicollinearity diagnostics results for the ESS data (Round 7).**

|        | VIF   | TOL   | Leamer | CVIF  |
|--------|-------|-------|--------|-------|
| HDI    | 3.202 | 0.312 | 0.559  | 3.363 |
| CPI    | 3.264 | 0.306 | 0.555  | 3.428 |
| PO     | 1.123 | 0.890 | 0.944  | 1.179 |
| CI     | 1.071 | 0.934 | 0.966  | 1.125 |
| PC     | 1.267 | 0.789 | 0.889  | 1.330 |
| TR     | 1.397 | 0.716 | 0.846  | 1.467 |
| ITR    | 2.207 | 0.453 | 0.673  | 2.318 |
| SAT    | 2.105 | 0.475 | 0.689  | 2.211 |
| AtI    | 1.714 | 0.583 | 0.764  | 1.800 |
| AtR    | 1.519 | 0.658 | 0.811  | 1.595 |
| PSR    | 1.622 | 0.617 | 0.785  | 1.704 |

which are Leamer's measure [50], and corrected variance inflation factor (CVIF) [51] for detecting the extent of multicollinearity in the data in line with previous research [48]. Similar to the TOLs, the Leamer's measure takes the values of between 0 and 1 for which the values close to zero indicate the existence of multicollinearity in the data. The CVIFs that are larger than 10 indicate a possible multicollinearity problem [49].

Table 5 displays these four measures obtained for the ESS (Round 7) data. Based on the results presented in this table, the VIFs for all the subject level variables are smaller than 3. The VIFs for aggregate variables HDI (i.e., 3.202 > 3) and CPI (i.e., 3.264 > 3) indicate a possible multicollinearity in the data. However, since other variables in the data have small VIFs and the VIFs for HDI and CPI are smaller than 5 (or 10) and not very large than 3, we conclude that the ESS data do not suffer from a severe multicollinearity problem. The results for other measures in Table 5 are in accordance with that for the VIF. As expected, the TOL s and Leamer's measures of the variables HDI and CPI are smaller than that of other variables in the data. However, these values are not too close to zero and the CVIFs for all variables in the model are quite smaller than the decision point 10 which indicates that there is no multicollinearity problem in the data.

Last but not least. It is important to mention that the continuous variables in the ESS data (Round 7) are standardized [7] to improve the interpretation of main effects in the presence of interactions. There are two main types of standardization techniques in the context of multi-level modeling: Standardization with grand mean centering and standardization with group mean centering. We standardized the continuous variables in the data using the most commonly used technique which is the standardization with grand mean centering in which the overall means of the variables are subtracted from their values and consequently divided by their standard deviations. Note that, we do not report the conditional effects of the individual level variables on the outcome at different values of the aggregate level variables in this study. The rationale behind this choice is that obtaining conditional effects is more valuable by applying standardization with group mean centering (stated otherwise within cluster centering) rather than grand mean centering in the presence of cross-level interactions [52, 53].

## General discussion

From demographics point of view, our results replicate previous research that shows males [54, 55], those less educated, and right-wing or right leaning individuals support populist parties more [56]. In terms of individual and country level processes, our findings also suggest that support for populist and thin ideology parties is driven by a fundamental intergroup

mechanism, i.e., social identity and status concerns, which in turn are influenced by country level processes, i.e., opportunities available for human development (HDI) and corruption perceptions (CPI). Below, we first discuss individual level processes. We then elaborate on our findings with regards to the direct effects of our country level variables HDI and CPI. Last but not least, we unpack our findings on cross–level interactions.

First and foremost, at the individual level. we did not find support for H1a, the positive effect of country identification on populist voting. Results showed that such an effect, if any is contingent on the levels of human development and corruption perceptions at the country level. While, previous research on identity and political behaviour argues that stronger identification with one's group is associated with political activism [57, 58], our results show that when the contents of that identity, in our case national identity, is perceived and experienced differently, this association is not straightforward. Our results, however, fully supported (H1b). Past research shows that dimensions of efficacy, i.e., participative versus group versus movement efficacy are associated with activism [59, 60]. These results replicate and extend findings from previous research in the sense that political confidence as a proxy of efficacy predicts populist support at the individual level.

We predicted that perceiving the political system as responsive, generalized and institutional trust, satisfaction with the system, and positive attitudes toward immigrants and refugees would be negatively associated with populist voting (H2a; H2b; H2c; H2d; H2e, and H2f). Previous research shows that trust in institutions [61] and satisfaction with the system [62] reduce support for populism. Our results replicate this finding. Most research to date investigated whether populist attitudes predict negative attitudes toward immigrants and refugees [63, 64], Our results confirm this finding in the opposite direction. We found that more positive attitudes toward refugees are associated with less support for populist parties. Cumulatively these findings confirm H2a-H2f.

## Moderating effects of country level variables on individual level variables

At the aggregate level, support for populist parties increases as the level of human development increases. As expected, populist voting also changes as a function of perceived corruption. The less corrupt the system is seen the less people support populist parties.

First, As HDI increases one attachment to her country, satisfaction with the system, and more importantly trust in the institutions positively predict support for populist parties. As the level of human development opportunities increases the null effect of country identification and the negative effect of satisfaction with the system as well as the negative effect of institutional trust on populist voting becomes positive. Taken together, we argue that this supports the "wealth paradox [22] but not the *losers of globalization thesis* [2, 16]. These findings contradict earlier findings at the individual level that populist support will be highest when individuals are not satisfied and when they do not trust the system [61, 62]. Future multilevel research using more specific measures of these variables, especially of country identification is welcome. In a similar vein, more complex models looking at the indirect effects of country identification via dimensions of trust, political confidence, or system satisfaction on populist voting might be able to reveal more about the impact of country identification on populist voting via alternative dimensions.

Second, the level of human development also moderates the effect of attitudes toward immigrants and support for policies benefitting the refugees. Initially negative, the effect of positive attitudes toward immigrants on populist voting disappears when we factor in the level of human development. In addition, whereas initially not significant the effect of attitudes toward refugees becomes significant. In more developed countries positive attitudes toward

refugee benefitting policies decrease support for populist parties. As our results show, there is a distinction between how people perceive immigrants and refugees. Perhaps, immigrants are perceived as more competent and thus threatening [65] than the refugees but incorporating aggregate level variables trumps this effect.

Third, our full model implies that corruption perceptions too moderate the effect of individual level variables on populist voting. Attachment to one's country for instance is associated with less support for populist parties when the system is seen as ethical and honest. In similar vein trust in other individuals negatively predicts populist support at the individual level. When, however, the system is perceived as ethical more generalized trust is associated with more support for populist parties. The trust in institutions, as expected, negatively predicts populist voting when the system is seen as ethical and fair. This perhaps provides a glimmer of hope towards combatting the populist surge. Earlier research shows that level of human development and corruption perceptions go hand in hand. Higher levels of human development are associated with lower level of corruption perceptions [66]. Measures toward emphasizing the ethical and honest nature of the system could outweigh the positively moderating effect of human development on populist voting.

## Limitations and implications

Comprehensiveness and novelty of our results notwithstanding, we acknowledge that we need to be cautious while interpreting the findings and drawing wider conclusions. First and foremost, we collapse left, centre, and right-wing parties and analyse them together. Recent research shows that support for populist parties across the political spectrum is motivated by alternative processes [67]. However, our focus on subjective experiences that are pervasive across the political spectrum, and operationalizing populism as a thin-centred ideology allows us to include all parties across the political spectrum. Accordingly, this helps us to identify a general set of psychological processes that are associated with populism irrespective of the ideological position. Second, we only use ESS Round 7 and provide a snapshot of the processes at a single point in time. Thus although comprehensive our results might not show the full scale of how populist attitudes changed over time. Third, our focus is limited in the sense that we do not provide a comparative analysis of our aggregate variables vis-à-vis traditional predictors (i.e., GDP per capita, Gini coefficient, net migration rate, or social welfare expenditure) established by previous research. Last but not least, we do not elaborate on country level differences. Taken together these limitations raise important questions that can be answered by future research. Longitudinal studies employing data from several rounds of ESS would be particularly welcome to compare *traditional* aggregate level predictors with relatively novel variables we include in our models. In a similar vein, case studies comparing countries that rank higher on HDI (e.g., Norway, Switzerland, or Iceland) with those ranking relatively lower (e.g., Hungary and Poland).

## Conclusion

In sum, these results provide important insights into how support for populism is a product of individual level social psychological processes, one's attachment to one's country, dimensions of trust, perceptions of the system, and one's satisfaction with the system. Perhaps, our most noteworthy finding is how two defining features of liberal economies, higher level of opportunities for human development within a system that is seen as ethical and honest, i.e., uncorrupted, influence and even reverse, rather paradoxically, the influence of some fundamental social psychological processes. Tellingly, the findings suggest that the support for populism, at least in our data, is likely to continue as the liberal democratic systems prosper but

governments and other institutional actors can mitigate the enhancing effects of this prosperity on populist voting.

## Supporting information

**S1 File.**
(DOCX)

## Author Contributions

**Conceptualization:** Hüseyin Çakal.

**Data curation:** Hüseyin Çakal.

**Formal analysis:** Yasin Altınışık.

**Funding acquisition:** Hüseyin Çakal.

**Methodology:** Yasin Altınışık.

**Project administration:** Hüseyin Çakal.

**Writing – original draft:** Hüseyin Çakal, Yasin Altınışık.

**Writing – review & editing:** Hüseyin Çakal, Ömer Gökcekuş, Ertugrul Gazi Eraslan.

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
