## [Decision Letter · Decision Letter 0]

19 May 2021

PONE-D-21-05815

Why People Vote for Thin-Centred Ideology Parties? A Multi-Level Multi-Country Test of Individual and Aggregate Level Predictors

PLOS ONE

Dear Dr. Cakal,

Thank you for submitting your manuscript to PLOS ONE. After careful consideration, we feel that it has merit but does not fully meet PLOS ONE’s publication criteria as it currently stands. Therefore, we invite you to submit a revised version of the manuscript that addresses the points raised during the review process.

The reviewers expressed positive interest in the manuscript you submitted but also raised some concerns about the paper. In particular, they both suggested improvements in the theoretical framework proposed and to give more details in analytic strategy, providing more foundation for reliability of the results.

We look forward to receiving your revised manuscript.

Kind regards,

Eleonora Brivio

Academic Editor

PLOS ONE

Journal Requirements:

2. Please amend your manuscript to include your abstract after the title page.

3. Please include your tables as part of your main manuscript and remove the individual files. Please note that supplementary tables (should remain/ be uploaded) as separate "supporting information" files

Reviewers' comments:

Reviewer's Responses to Questions

**Comments to the Author**

1. Is the manuscript technically sound, and do the data support the conclusions?

Reviewer #1: Partly

Reviewer #2: Yes

2. Has the statistical analysis been performed appropriately and rigorously? 

Reviewer #1: No

Reviewer #2: Yes

3. Have the authors made all data underlying the findings in their manuscript fully available?

Reviewer #1: Yes

Reviewer #2: Yes

4. Is the manuscript presented in an intelligible fashion and written in standard English?

Reviewer #1: Yes

Reviewer #2: Yes

5. Review Comments to the Author

Reviewer #1: The paper presents a multi-level analysis of country-wide and individual predictors of vote for populist parties in 23 European countries.

I appreciated the Authors' approach to the phenomenon and their attempt to bring together an individual-based focus, typical of social psychological research, and a focus on aggragate-level cross-national differences, typical of comparative political science research. I also found some of the results quite interesting in this light, as they suggest that different macro-social context can change and even reverse the effects of individual variables.

I also have some reservations on the paper, based on two main points, namely the theoretical framing of the introduction, which is not always consistent, and the design and description of the analyses, which should be checked to make sure that it complies with sufficient statistical standards. You can find below my observations on these two main points.

As for the first issue, the Authors' argument starts from two well established theories on the roots of support for populist movements, the economic anxiety and cultural backlash hypotheses, and then moves on to the wealth paradox hypothesis, which combines some elements of the cultural backlash hypothesis (the prevalence of racism and anti-immigrant attitudes) with an economic explanation, attributing it to prospective fear of a diminished status, rather than actual impoverishment. The following paragraph (Ideational approach to populism), however, deviates from this framework, introducing different factors (education at the societal level, and entitlement and empowerment at the individual level), and discussing the role of the crucial element of populist leaders and their rhetoric. The effect of this factor is not discussed directly, but rather connected with the idea of the "corrupted élite" argument enabling citizens' group categorisation (people-ingroup vs. élite-outgroup) and their distrust in politicians and the political system. This connection appears less linear than the preceding ones, and should be revised and more explicitly supported theoretically, for instance by referring to past research on social identity in political and collective action, an area the Authors seem to be familiar with, given the work they reference. Several other constructs are introduced in the latter part of this paragraph (p. 9), some of which probably deserved a more thorough discussion, such as the concept of country identification, which has been already investigated in relation to populism (e.g., see Marchlewska et al., 2018).

When they came to formulate specific hypotheses for their study, the Author chose to mix together some of the elements of the previously established models/hypotheses, making two new hypotheses apparently based on the valence of the predicted relationship with populist vote (positive for H1, negative for H2), and a separate hypothesis (H3) regarding the aggregate-level expected predictors of populist vote. They then combine these hypotheses into a moderation hypothesis (H4), in which they only state that the effects predicted in H1-3 would be "enhanced". From the subsequent elaboration (p. 11, l. 244-252) it seems that they expect to find stronger effects of the predictors used in H1&2 in the conditions which, according to H3, would lead to greater support for populist parties, but it is not completely clear if they expect the effects to become weaker or reverse in the opposite conditions (e.g., with low HDI and high CPI). This hypothesis should be more thoroughly discussed and explained. I am not fully convinced of the former three hypotheses, either, as I think it would be more appropriate to base them on existing theories and hypotheses from the literature (e.g., the economic anxiety and cultural backlash hypotheses), rather than regrouping variables. But perhaps the Authors can provide a convincing rationale for their choice.

My second reservation concerns the multi-level analyses reported in the results. I concur with the Authors that this is the appropriate approach to investigate the interplay between individual-level and aggregate-level variables in promoting vote for populist parties, and I appreciated their work in establishing the appropriateness of the models they tested (see the preliminary analyses reported on p. 15-16 and in the supplementary materials). What I am concerned about, however, is the reliability of the results, given the number of variables included in the model and the unclear links among them. I suspect that the models might have a multicollinearity problem, as some variables are probably strongly correlated with each other. The Authors should report multicollinarity diagnostic indexes (some are proabably available in the statistical packages used to run the analyses) to exclude or at least quantify the problem in their results, and help readers in their interpretation. Perhaps some basic zero-order correlations among the predictors could be reported in Table 1, as well. Furthermore, the description of the moderation effects (pp. 17-20) is not always clear, for instance when they say that when the moderator is "taken into account" the effects of the other variables change or do not change. A more typical approach would have been to report conditional effects of the focal predictors (the individual-level ones) at different levels of the moderators (the aggregate-level variables), but again the Authors may provide an explanation and justification for their choice.

All things considered, I found the paper interesting and I appreciated its goal of bringing together two different approaches to the object of analysis, so I hope the Authors will be able to make the necessary improvements to provide theoretically sound and statistically convincing support for their endeavour.

Reviewer #2: Overall, I enjoyed reading the manuscript, I though it drew on interesting individual and aggregate level data and there are some really interesting findings in there about populist support in general (as a phenomenon) regardless of ideological leaning. That being said, there is also some concern about this approach – as Mudde argues with the use of thin-centred ideology for defining populism, populism only becomes contextualized and given meaning when it becomes anchored within a thick ideology (which will differ in left- and right-wing contexts). I would like to have seen more discussion in the introduction of the ideational approach as to how mapping out these trends in the data, without contextualizing populism as either right or left, is justified. I think the logic is there to some degree, the emphasis on understanding the characteristics and links to social psych and structural-level phenomena, but this could be clarified a bit more. Similarly, by the time the reader gets to the discussion, this has been somewhat forgotten, and a reminder would be useful.

The paper outlines a series of social psychological processes that are said to fuel populist support (including identification, trust, system satisfaction, attitudes towards immigrants, efficacy and structural factors) yet these are discussed more in a narrative form and less in a way that links them theoretically. While this might not be necessary, an overarching ‘model’ or framework that positions these phenomena within an identity/hierarchy and system framework could be useful. Something similar to this has been recently outlined in the following paper ( https://doi.org/10.1016/j.copsyc.2020.06.009 ) which draws on many of the same concepts, papers and ideas developed here, but tries to position them within a theoretical framework of sorts. In particular, the discussion on hierarchy and status concerns seems relevant here, as it is not only that identities are mobilized as ingroup and outgroup, but also that these are positioned in ways that are threatening and perceived as unjust (relative deprivation findings), which subsequently align both low-income and high-income earners within the same ‘ingroup’ because it is not only about economic anxieites, as you point out, but also about the status concerns they express. Furthermore, the paper makes reference to Brubaker’s distinction between horizontal and vertical opposition which might be useful in accounting for how the ingroup is both defined against a horizontal other (corrupt elites) and vertical other (migrants/ refugees) in contexts of right-wing populism. Having a look at these references might help develop a sharper framing of the various items used, rather than stating that these are taken from different literatures and brought together. Mainly, I make this point to strengthen the overall rationale and framing for how these individual level and structural level processes interplay, and for the hypotheses generated which are sound.

Lastly, I think the discussion needs a bit of work. As mentioned above, there are some contradictory trends in the data when the aggregate level analysis is included, and this needs to be spelled out a bit more in detail. The discussion feels like it mainly repeats the findings but doesn’t move beyond that to consider more in-depth the implications (theoretically and practically) of these novel findings. This is mainly a push for the authors to really consider why these findings emerged and what they say about understandings of populism. This is nicely done in relation to trust, but could be expanded on for other processes discussed. Perhaps tightening the framework that links the different processes in the introduction helps to revise this part.

Minor points:

A re-read of the abstract for typos, missing punctuation and missing words such as ‘a’ ,’of’ etc. is needed. I have listed a few of these below, but not all.

Abstract, typo:

Thus, we provide the first evidence that the populist surge is a product of a complex set (of) social psychological mechanisms that are moderated by the general level of development and corruption perceptions in a country

P2, line 31 – extra “ after ideology: of populism as a “thin-centred” ideology”

P2, line 34: ). Liberal democracies on the other hand (stating ‘on the other hand’ assumes there’s a ‘on the one hand’ prior in the text, which doesn’t appear, so I’d suggest revising this)

P2 , line 48: the level (of) human development and corruption

P3, line 69 – In (a) similar vein, (the) cultural backlash…

P 7, line 145 - Although crucial to mobilization (for mobilizing? / to the mobilization of?) this collective identity,

6. PLOS authors have the option to publish the peer review history of their article (what does this mean?). If published, this will include your full peer review and any attached files.

Reviewer #1: **Yes: **Mauro Bertolotti

Reviewer #2: No

---

## [Author Response · Author response to Decision Letter 0]

1 Dec 2021

Response to the reviewers

Dear Editor,

Thank you very much for the detailed feedback and for the supportive editorial process. Upfront please accept our sincere apologies for not completing this revision earlier. The loss of a close family member and health related problems prevented the lead author from engaging with and completing the review. 

We think we did our best to integrate the feedback into the manuscript. Below, you can find our detailed responses. Changes are highlighted as track changes. We numbered each comment and provided our responses accordingly. 

Once again, allow us to express our gratitude for the supportive editorial process and the insightful comments which made the manuscript stronger. Please do not hesitate to contact us if you have further comments. 

Sincerely,

Huseyin Cakal

P.S. We were not aware of the journal’s requirement of completing the review within 6 months of receiving editorial feedback. This is of course not an excuse, but we want to reiterate our commitment to revise the manuscript again should the reviewers consider it necessary.

R1-1 

R1 comments that “the Authors' argument starts from two well established theories on the roots of support for populist movements, the economic anxiety and cultural backlash hypotheses, and then moves on to the wealth paradox hypothesis, which combines some elements of the cultural backlash hypothesis (the prevalence of racism and anti-immigrant attitudes) with an economic explanation, attributing it to prospective fear of a diminished status, rather than actual impoverishment. The following paragraph (Ideational approach to populism), however, deviates from this framework, introducing different factors (education at the societal level, and entitlement and empowerment at the individual level), and discussing the role of the crucial element of populist leaders and their rhetoric. The effect of this factor is not discussed directly, but rather connected with the idea of the "corrupted élite" argument enabling citizens' group categorisation (people-ingroup vs. élite-outgroup) and their distrust in politicians and the political system. This connection appears less linear than the preceding ones, and should be revised and more explicitly supported theoretically, for instance by referring to past research on social identity in political and collective action, an area the Authors seem to be familiar with, given the work they reference. Several other constructs are introduced in the latter part of this paragraph (p. 9), some of which probably deserved a more thorough discussion, such as the concept of country identification, which has been already investigated in relation to populism (e.g., see Marchlewska et al., 2018).”

We thank R1 for these insightful and useful comments. We have now incorporated these ideas and R2 comments on the same issue to the manuscript. More specifically, we have changed our framing our populism in line with Brubaker’s vertical vs horizontal perspective and incorporated Obradovic et al’s argument to our theorizing. In addition, we dropped our emphasis on ideational approach to populism. 

R2-2 R1 also comments that “When they came to formulate specific hypotheses for their study, the Author chose to mix together some of the elements of the previously established models/hypotheses, making two new hypotheses apparently based on the valence of the predicted relationship with populist vote (positive for H1, negative for H2), and a separate hypothesis (H3) regarding the aggregate-level expected predictors of populist vote. They then combine these hypotheses into a moderation hypothesis (H4), in which they only state that the effects predicted in H1-3 would be "enhanced". From the subsequent elaboration (p. 11, l. 244-252) it seems that they expect to find stronger effects of the predictors used in H1&2 in the conditions which, according to H3, would lead to greater support for populist parties, but it is not completely clear if they expect the effects to become weaker or reverse in the opposite conditions (e.g., with low HDI and high CPI). This hypothesis should be more thoroughly discussed and explained. I am not fully convinced of the former three hypotheses, either, as I think it would be more appropriate to base them on existing theories and hypotheses from the literature (e.g., the economic anxiety and cultural backlash hypotheses), rather than regrouping variables. But perhaps the Authors can provide a convincing rationale for their choice.”

We have now completely revised our hypotheses and re-arranged them in line with our introduction (pp8-10). We think this rearranged format provides clarity but we are happy to revise again if the RRs consider it necessary. 

R1-3. R1 comments that “My second reservation concerns the multi-level analyses reported in the results. I concur with the Authors that this is the appropriate approach to investigate the interplay between individual-level and aggregate-level variables in promoting vote for populist parties, and I appreciated their work in establishing the appropriateness of the models they tested (see the preliminary analyses reported on p. 15-16 and in the supplementary materials). What I am concerned about, however, is the reliability of the results, given the number of variables included in the model and the unclear links among them. I suspect that the models might have a multicollinearity problem, as some variables are probably strongly correlated with each other. The Authors should report multicollinarity diagnostic indexes (some are proabably available in the statistical packages used to run the analyses) to exclude or at least quantify the problem in their results, and help readers in their interpretation. Perhaps some basic zero-order correlations among the predictors could be reported in Table 1, as well. Furthermore, the description of the moderation effects (pp. 17-20) is not always clear, for instance when they say that when the moderator is "taken into account" the effects of the other variables change or do not change.”

We have now incorporate additional statistics on multicollinearity and centering (pp18-20)- and descriptive statistics p 13) to help ease interpretation of the results. 

R1-4

And adds that “A more typical approach would have been to report conditional effects of the focal predictors (the individual-level ones) at different levels of the moderators (the aggregate-level variables), but again the Authors may provide an explanation and justification for their choice.”

We are not aware of such a technical capability in R, our choice of software or theoretical work that discusses obtaining the conditional effects of focal predictors at different level of moderators in multilevel models. However, we are more than happy to re-consider this option if the R1 could provide some details. 

All things considered, I found the paper interesting and I appreciated its goal of bringing together two different approaches to the object of analysis, so I hope the Authors will be able to make the necessary improvements to provide theoretically sound and statistically convincing support for their endeavour.

We are most grateful for R1’s positive approach and insightful comments. 

R2-1

R2 contends that “I would like to have seen more discussion in the introduction of the ideational approach as to how mapping out these trends in the data, without contextualizing populism as either right or left, is justified. I think the logic is there to some degree, the emphasis on understanding the characteristics and links to social psych and structural-level phenomena, but this could be clarified a bit more. Similarly, by the time the reader gets to the discussion, this has been somewhat forgotten, and a reminder would be useful”

This is indeed one of key issues that we struggled with most. However, after careful consideration and in line with both R1’s comments above and R2’s further comments below we decide the employ Brubaker’s 2 dimensional framework on populism. We believe this approach fits nicely with our aim to analyse populist voting across both ends of the political spectrum. We elaborate on this more below. .

R2-2

R2 also comments that “The paper outlines a series of social psychological processes that are said to fuel populist support (including identification, trust, system satisfaction, attitudes towards immigrants, efficacy and structural factors) yet these are discussed more in a narrative form and less in a way that links them theoretically. While this might not be necessary, an overarching ‘model’ or framework that positions these phenomena within an identity/hierarchy and system framework could be useful. Something similar to this has been recently outlined in the following paper (https://doi.org/10.1016/j.copsyc.2020.06.009) which draws on many of the same concepts, papers and ideas developed here, but tries to position them within a theoretical framework of sorts”

We are most grateful to R2 for pointing this work and Brubaker’s work to us. In our readings, we have completely missed this line of work. We now believe we have incorporated this comment by emphasizing this in our introduction (pp1-2 & pp4 -5). 

R2-3

R3 also adds that “ In particular, the discussion on hierarchy and status concerns seems relevant here, as it is not only that identities are mobilized as ingroup and outgroup, but also that these are positioned in ways that are threatening and perceived as unjust (relative deprivation findings), which subsequently align both low-income and high-income earners within the same ‘ingroup’ because it is not only about economic anxieites, as you point out, but also about the status concerns they express. Furthermore, the paper makes reference to Brubaker’s distinction between horizontal and vertical opposition which might be useful in accounting for how the ingroup is both defined against a horizontal other (corrupt elites) and vertical other (migrants/ refugees) in contexts of right-wing populism”

As per our response to R2-2 comments we have erroneously ignored this line of work. We now refer to Brubaker’s 2 dimensional model as well as to the arguments put forward by Obradovic et al. 

Also 

“ Having a look at these references might help develop a sharper framing of the various items used, rather than stating that these are taken from different literatures and brought together. Mainly, I make this point to strengthen the overall rationale and framing for how these individual level and structural level processes interplay, and for the hypotheses generated which are sound”.

We are most grateful for these positive comments. Although, we did our best to incorporate them we might have still fallen short of R3 expectations. We are therefore happy to consider additional guidance on this. 

R2-4

“R2 comments that “Lastly, I think the discussion needs a bit of work. As mentioned above, there are some contradictory trends in the data when the aggregate level analysis is included, and this needs to be spelled out a bit more in detail. The discussion feels like it mainly repeats the findings but doesn’t move beyond that to consider more in-depth the implications (theoretically and practically) of these novel findings. This is mainly a push for the authors to really consider why these findings emerged and what they say about understandings of populism. This is nicely done in relation to trust, but could be expanded on for other processes discussed. Perhaps tightening the framework that links the different processes in the introduction helps to revise this part.”

Once again, these are very helpful. We did our best to rework our discussion and reinforced with additional discussion on how our findings contradict or align with previous research. 

Minor points:

A re-read of the abstract for typos, missing punctuation and missing words such as ‘a’ ,’of’ etc. is needed. I have listed a few of these below, but not all.

Abstract, typo:

Thus, we provide the first evidence that the populist surge is a product of a complex set (of) social psychological mechanisms that are moderated by the general level of development and corruption perceptions in a country

P2, line 31 – extra “ after ideology: of populism as a “thin-centred” ideology”

P2, line 34: ). Liberal democracies on the other hand (stating ‘on the other hand’ assumes there’s a ‘on the one hand’ prior in the text, which doesn’t appear, so I’d suggest revising this)

P2 , line 48: the level (of) human development and corruption

P3, line 69 – In (a) similar vein, (the) cultural backlash…

P 7, line 145 - Although crucial to mobilization (for mobilizing? / to the mobilization of?) this collective identity,

We have now corrected these typos.

---

## [Decision Letter · Decision Letter 1]

26 Jan 2022

PONE-D-21-05815R1Why People Vote for Thin-Centred Ideology Parties? A Multi-Level Multi-Country Test of Individual and Aggregate Level PredictorsPLOS ONE

Dear Dr. Cakal,

Thank you for submitting your manuscript to PLOS ONE. After careful consideration, we feel that it has merit but does not fully meet PLOS ONE’s publication criteria as it currently stands. Therefore, we invite you to submit a revised version of the manuscript that addresses the points raised during the review process.

We look forward to receiving your revised manuscript.

Kind regards,

Ghaffar Ali, PhD

Academic Editor

PLOS ONE

Journal Requirements:

Reviewers' comments:

Reviewer's Responses to Questions

**Comments to the Author**

1. If the authors have adequately addressed your comments raised in a previous round of review and you feel that this manuscript is now acceptable for publication, you may indicate that here to bypass the “Comments to the Author” section, enter your conflict of interest statement in the “Confidential to Editor” section, and submit your "Accept" recommendation.

Reviewer #2: All comments have been addressed

2. Is the manuscript technically sound, and do the data support the conclusions?

Reviewer #2: Yes

3. Has the statistical analysis been performed appropriately and rigorously? 

Reviewer #2: No

4. Have the authors made all data underlying the findings in their manuscript fully available?

Reviewer #2: Yes

5. Is the manuscript presented in an intelligible fashion and written in standard English?

Reviewer #2: Yes

6. Review Comments to the Author

Reviewer #2: I am happy with how the revised manuscript addresses my comments; the revisions to the framing of the paper allow for more coherence and clarity as to what it is doing, and it helps also in allowing for the discussion to be explicit about its contributions.

I would only suggest that the authors proofread the changes made. In reading the tracked changes version of the document I came across a number of instances where proof-reading was required. i include only two examples below from the Discussion, but i recommend the authors still read eveyrthing over (or have someone else do so - sometimes we become quite blind to these things when we've been working on a manuscript for a long time!)

"by a fundamental intergroup mechanisms, i.e. identity and status concerns which in turn influenced by country level processes" - 'which in turn ARE influenced by'

"Results showed that such an effect, if any is contingent to the levels of human development and corruption perceptions at the country level." ''Results showed that such an effect, if any, is contingent to... (usually it's contingent on?')'

7. PLOS authors have the option to publish the peer review history of their article (what does this mean?). If published, this will include your full peer review and any attached files.

Reviewer #2: No

---

## [Author Response · Author response to Decision Letter 1]

28 Jan 2022

R1 

R2 comments that I am happy with how the revised manuscript addresses my comments; the revisions to the framing of the paper allow for more coherence and clarity as to what it is doing, and it helps also in allowing for the discussion to be explicit about its contributions”

We are most grateful for these encouraging words. 

R2 further comments that “I would only suggest that the authors proofread the changes made. In reading the tracked changes version of the document I came across a number of instances where proof-reading was required. i include only two examples below from the Discussion, but i recommend the authors still read eveyrthing over (or have someone else do so - sometimes we become quite blind to these things when we've been working on a manuscript for a long time!)

"by a fundamental intergroup mechanisms, i.e. identity and status concerns which in turn influenced by country level processes" - 'which in turn ARE influenced by'

"Results showed that such an effect, if any is contingent to the levels of human development and corruption perceptions at the country level." ''Results showed that such an effect, if any, is contingent to... (usually it's contingent on?')”

We have proofread the ms and further improved the overall style & edited in these suggestions.

---

## [Decision Letter · Decision Letter 2]

11 Feb 2022

Why People Vote for Thin-Centred Ideology Parties? A Multi-Level Multi-Country Test of Individual and Aggregate Level Predictors

PONE-D-21-05815R2

Dear Dr. Cakal,

We’re pleased to inform you that your manuscript has been judged scientifically suitable for publication and will be formally accepted for publication once it meets all outstanding technical requirements.

Kind regards,

Ghaffar Ali, PhD

Academic Editor

PLOS ONE

Additional Editor Comments (optional):

Reviewers' comments:

Reviewer's Responses to Questions

**Comments to the Author**

1. If the authors have adequately addressed your comments raised in a previous round of review and you feel that this manuscript is now acceptable for publication, you may indicate that here to bypass the “Comments to the Author” section, enter your conflict of interest statement in the “Confidential to Editor” section, and submit your "Accept" recommendation.

Reviewer #2: (No Response)

2. Is the manuscript technically sound, and do the data support the conclusions?

Reviewer #2: Yes

3. Has the statistical analysis been performed appropriately and rigorously? 

Reviewer #2: Yes

4. Have the authors made all data underlying the findings in their manuscript fully available?

Reviewer #2: Yes

5. Is the manuscript presented in an intelligible fashion and written in standard English?

Reviewer #2: Yes

6. Review Comments to the Author

Reviewer #2: Congratulations to the authors for improving the manuscript overall. i very much enjoyed the paper and i think others will too.

7. PLOS authors have the option to publish the peer review history of their article (what does this mean?). If published, this will include your full peer review and any attached files.

Reviewer #2: No

---

## [Editor Report · Acceptance letter]

21 Feb 2022

PONE-D-21-05815R2 

Why People Vote for Thin-Centred Ideology Parties? A Multi-Level Multi-Country Test of Individual and Aggregate Level Predictors 

Dear Dr. Çakal:

I'm pleased to inform you that your manuscript has been deemed suitable for publication in PLOS ONE. Congratulations! Your manuscript is now with our production department. 

Kind regards, 

on behalf of

Prof. Ghaffar Ali 

Academic Editor

PLOS ONE